# Analytical validity of nanopore sequencing for rapid SARS-CoV-2 genome analysis

Rowena A. Bull[1,2,13], Thiruni N. Adikari[1,2,13], James M. Ferguson [3], Jillian M. Hammond [3], Igor Stevanovski [3], Alicia G. Beukers[4], Zin Naing[2,5], Malinna Yeang[2,5], Andrey Verich[1], Hasindu Gamaarachchi [3,6], Ki Wook Kim[5,7], Fabio Luciani[1,2], Sacha Stelzer-Braid[2,5], John-Sebastian Eden [8,9], William D. Rawlinson[2,5,7,10], Sebastiaan J. van Hal [4,11] & Ira W. Deveson [3,12,13 ✉]

Viral whole-genome sequencing (WGS) provides critical insight into the transmission and evolution of Severe Acute Respiratory Syndrome Coronavirus 2 (SARS-CoV-2). Long-read sequencing devices from Oxford Nanopore Technologies (ONT) promise significant improvements in turnaround time, portability and cost, compared to established short-read sequencing platforms for viral WGS (e.g., Illumina). However, adoption of ONT sequencing for SARS-CoV-2 surveillance has been limited due to common concerns around sequencing accuracy. To address this, here we perform viral WGS with ONT and Illumina platforms on 157 matched SARS-CoV-2-positive patient specimens and synthetic RNA controls, enabling rigorous evaluation of analytical performance. We report that, despite the elevated error rates observed in ONT sequencing reads, highly accurate consensus-level sequence determination was achieved, with single nucleotide variants (SNVs) detected at >99% sensitivity and >99% precision above a minimum ~60-fold coverage depth, thereby ensuring suitability for SARS-CoV-2 genome analysis. ONT sequencing also identified a surprising diversity of structural variation within SARS-CoV-2 specimens that were supported by evidence from short-read sequencing on matched samples. However, ONT sequencing failed to accurately detect short indels and variants at low read-count frequencies. This systematic evaluation of analytical performance for SARS-CoV-2 WGS will facilitate widespread adoption of ONT sequencing within local, national and international COVID-19 public health initiatives.

[1] The Kirby Institute for Infection and Immunity, University of New South Wales, Sydney, NSW, Australia. [2] School of Medical Sciences, Faculty of Medicine, University of New South Wales, Sydney, NSW, Australia. [3] Kinghorn Centre for Clinical Genomics, Garvan Institute of Medical Research, Sydney, NSW, Australia. [4] NSW Health Pathology, Department of Infectious Diseases and Microbiology, Royal Prince Alfred Hospital, Sydney, NSW, Australia. [5] Virology Research Laboratory, Serology and Virology Division (SAViD), NSW Health Pathology, Prince of Wales Hospital, Sydney, NSW, Australia. [6] School of Computer Science and Engineering, University of New South Wales, Sydney, NSW, Australia. [7] School of Women's and Children's Health, Faculty of Medicine, University of New South Wales, Sydney, NSW, Australia. [8] Marie Bashir Institute for Infectious Diseases and Biosecurity & Sydney Medical School, The University of Sydney, Sydney, NSW, Australia. [9] Centre for Virus Research, Westmead Institute for Medical Research, Sydney, NSW, Australia. [10] School of Biotechnology and Biomolecular Sciences, Faculty of Science, University of New South Wales, Sydney, NSW, Australia. [11] Central Clinical School, University of Sydney, Sydney, NSW, Australia. [12] St Vincent's Clinical School, Faculty of Medicine, University of New South Wales, Sydney, NSW, Australia. [13]These authors contributed equally: Rowena A. Bull, Thiruni N. Adikari, Ira W. Deveson. ✉email: i.deveson@garvan.org.au

Severe acute respiratory syndrome coronavirus 2 (SARS-CoV-2) is the causative pathogen for COVID-19 disease[1,2]. SARS-CoV-2 is a positive-sense single-stranded RNA virus with a ~30-kb poly-adenylated genome[1,2]. Complete genome sequences published in January 2020[1,3] enabled development of RT-PCR assays for SARS-CoV-2 detection that have served as the diagnostic standard during the ongoing COVID-19 pandemic[4].

Whole-genome sequencing (WGS) of SARS-CoV-2 provides additional data to complement routine diagnostic testing. Viral WGS informs public health responses by defining the phylogenetic structure of disease outbreaks[5]. Integration with epidemiological data identifies transmission networks and can infer the origin of unknown cases[6–11]. Largescale, longitudinal surveillance by viral WGS may also provide insights into virus evolution, with important implications for vaccine development[12–15].

WGS can be performed via PCR amplification or hybrid-capture of the reverse-transcribed SARS-CoV-2 genome sequence, followed by high-throughput sequencing. Short-read sequencing technologies (e.g., Illumina) enable accurate sequence determination and are the current standard for pathogen genomics. However, long-read sequencing devices from Oxford Nanopore Technologies (ONT) offer an alternative with several advantages. ONT devices are portable, cheap, require minimal supporting laboratory infrastructure or technical expertise for sample preparation, and can be used to perform rapid sequencing analysis with flexible scalability[16].

The use of ONT devices for viral surveillance has been demonstrated during Ebola, Zika and other disease outbreaks[17–19]. Although protocols for ONT sequencing of SARS-CoV-2 have been established and applied in both research and public health settings[20–22], adoption of the technology has been limited due to concerns around accuracy. ONT devices exhibit lower read-level sequencing accuracy than short-read platforms[23–25]. This may have a disproportionate impact on SARS-CoV-2 analysis, due to the virus' low mutation rate ($8 \times 10^{-4}$ substitutions per site per year[26]), which ensures erroneous (false-positive) or undetected (false-negative) genetic variants have a strong confounding effect.

In order to address concerns regarding ONT sequencing accuracy and evaluate its analytical validity for SARS-CoV-2 genomics, we have performed amplicon-based nanopore and short-read WGS on matched SARS-CoV-2-positive patient specimens and synthetic RNA controls, allowing rigorous evaluation of ONT performance characteristics.

## Results

**Analysis of synthetic SARS-CoV-2 controls.** Synthetic DNA or RNA reference standards can be used to assess the accuracy and reproducibility of next-generation sequencing assays[27]. We first sequenced synthetic RNA controls that were generated by in vitro transcription of the SARS-CoV-2 genome sequence. The controls matched the Wuhan-Hu-1 reference strain at all positions, allowing analytical errors to be unambiguously identified. To mimic a real-world viral WGS experiment, synthetic RNA was reverse-transcribed then amplified using multiplexed PCR of $98 \times ~400$ bp amplicons that enabled evaluation of ~95% of the SARS-CoV-2 genome. Eight independent replicates were sequenced on ONT PromethION and Illumina MiSeq instruments (see Methods section).

We aligned the resulting reads to the Wuhan-Hu-1 reference genome to assess sequencing accuracy and related quality metrics (Supplementary Fig. 1a–i). Illumina and ONT platforms exhibited distinct read-level error profiles, with the latter characterised by an elevated rate of both substitution (23-fold) and insertion-deletion (indel) errors (76-fold; Table 1 and Supplementary Fig. 1d, e). Per-base error frequency profiles

showed clear correlation between ONT replicates (substitution $R^2 = 0.67$; indel $R^2 = 0.82$; Supplementary Fig. 1f, g). This indicates that ONT sequencing errors are not entirely random but are influenced by local sequence context. For example, indel errors were enriched (1.4-fold) at low-complexity sequences within the SARS-CoV-2 genome (i.e., sites with homopolymeric or repetitive content; ~1% of the genome; Supplementary Fig. 1d). Illumina error profiles showed weaker correlation between replicates (substitution $R^2 = 0.15$; indel $R^2 = 0.42$), indicating that short-read sequencing errors were less systematic than for ONT libraries (Supplementary Fig. 1h, i).

Despite their distinct error profiles, both sequencing platforms demonstrated high consensus-level sequencing accuracy across the SARS-CoV-2 genome. We used *iVar* and *Medaka* workflows to determine consensus genome sequences for Illumina and ONT libraries, respectively (see Methods section). We detected just two erroneous variant candidates in a single ONT library (Table 1). Both of these were single-base insertions occurring at low-complexity sites (Supplementary Fig. 2), with no erroneous SNVs detected in any replicate ($n = 8$). All Illumina libraries exhibited perfect accuracy (Table 1). Therefore, the sequencing artefacts affecting both technologies had minimal impact on the accuracy of consensus-level sequence determination, with indel errors in ONT samples being a possible exception.

**Analysis of matched patient isolates.** To further evaluate the suitability of ONT sequencing for SARS-CoV-2 genomics, we conducted rigorous proficiency testing using bona fide clinical specimens. We performed ONT and Illumina WGS on matched, de-identified SARS-CoV-2-positive cases collected at public hospital laboratories in Eastern & Southern New South Wales and Metropolitan Sydney from March to April 2020 (see Methods section). Selected specimens covered a range of SARS-CoV-2 lineages and viral titres (~$10^3$–$10^8$ copies/μL; Supplementary Data 1). The SARS-CoV-2 genome was enriched by PCR amplification, using a custom set of $14 \times ~2.5$ kb amplicons that covers 29,783/29,903 bp (99.6%) of the genome, including 100% of annotated protein-coding positions[6]. Pooled amplicons then underwent parallel library preparation and sequencing on an ONT GridION/PromethION and an Illumina MiSeq instrument (see Methods section). Short-read sequencing was performed according to a pathogen genomics accredited diagnostic workflow in a reference NSW Health Pathology laboratory, enabling direct comparison of nanopore sequencing to the established standard for pathogen genomics.

In total, we obtained complete (99.6%) genome coverage with both technologies for 157 matched positive cases (Supplementary Data 1). By comparison to the Wuhan-Hu-1 reference strain, Illumina sequencing identified 7.6 consensus single-nucleotide variants (SNVs) and 0.04 indels, on average, per sample. A further 1.0 SNVs and 0.2 indels per sample were detected at sub-consensus read-count frequencies (20–80%), indicative of intra-specimen genetic diversity (see below). Excluding positions with evidence of sub-consensus variation, this provides an overall comparison set of 1201 consensus variants and 4,674,554 positions that match the reference strain in a given sample, against which to assess the accuracy of SARS-CoV-2 nanopore sequencing (Supplementary Data 1).

We used each of two best-practice bioinformatics pipelines developed by the ARTIC network to identify consensus variants with ONT sequencing data. The alternative pipelines differed primarily in their use of either *Medaka* or *Nanopolish* to call variants (see Methods section). In general, ONT variant candidates identified by both pipelines were highly concordant with the Illumina comparison set. Illumina variants were detected

**Table 1 Sequencing accuracy for Illumina and ONT whole-genome sequencing of synthetic SARS-CoV-2 controls.**

| | | Read-level error rate (errors per base per read) | | | | Erroneous variants | | | |
|---|---|---|---|---|---|---|---|---|---|
| Illumina samples | Reportable (bp) | Total | Mismatch | Deletion | Insertion | Total | SNVs | Indels | Consensus accuracy (%) |
| A | 28,687 | 0.00152 | 0.00083 | 0.00058 | 0.00011 | 0 | 0 | 0 | 100 |
| B | 28,687 | 0.00153 | 0.00082 | 0.00060 | 0.00012 | 0 | 0 | 0 | 100 |
| C | 28,687 | 0.00148 | 0.00079 | 0.00057 | 0.00012 | 0 | 0 | 0 | 100 |
| D | 28,687 | 0.00172 | 0.00098 | 0.00063 | 0.00011 | 0 | 0 | 0 | 100 |
| E | 28,687 | 0.00124 | 0.00089 | 0.00024 | 0.00011 | 0 | 0 | 0 | 100 |
| F | 28,687 | 0.00170 | 0.00137 | 0.00023 | 0.00011 | 0 | 0 | 0 | 100 |
| G | 28,687 | 0.00122 | 0.00088 | 0.00022 | 0.00011 | 0 | 0 | 0 | 100 |
| H | 28,687 | 0.00118 | 0.00084 | 0.00024 | 0.00011 | 0 | 0 | 0 | 100 |
| Mean | 28,687 | 0.00145 | 0.00092 | 0.00041 | 0.00011 | 0 | 0 | 0 | 100 |
| **ONT samples** | **Reportable (bp)** | **Total** | **Mismatch** | **Deletion** | **Insertion** | **Total** | **SNVs** | **Indels** | **Consensus accuracy (%)** |
| A | 28,192 | 0.06067 | 0.02093 | 0.02475 | 0.01499 | 0 | 0 | 0 | 100 |
| B | 28,192 | 0.06180 | 0.02150 | 0.02527 | 0.01503 | 0 | 0 | 0 | 100 |
| C | 28,192 | 0.06114 | 0.02141 | 0.02476 | 0.01496 | 0 | 0 | 0 | 100 |
| D | 28,192 | 0.06110 | 0.02146 | 0.02471 | 0.01493 | 0 | 0 | 0 | 100 |
| E | 28,192 | 0.06013 | 0.02067 | 0.02445 | 0.01501 | 0 | 0 | 0 | 100 |
| F | 28,192 | 0.05972 | 0.02018 | 0.02457 | 0.01496 | 2 | 0 | 2 | 99.9929 |
| G | 28,192 | 0.06178 | 0.02173 | 0.02486 | 0.01520 | 0 | 0 | 0 | 100 |
| H | 28,192 | 0.06030 | 0.02049 | 0.02470 | 0.01511 | 0 | 0 | 0 | 100 |
| Mean | 28,192 | 0.06083 | 0.02105 | 0.02476 | 0.01502 | 0.25 | 0 | 0.25 | 99.9991 |

**Table 2 Consensus-level accuracy of ONT whole-genome SARS-CoV-2 sequencing on patient specimens.**

| | Medaka | Medaka minus blacklist[a] | Nanopolish | Nanopolish minus blacklist[a] |
|---|---|---|---|---|
| Cases analysed | 157 | 157 | 157 | 157 |
| Genome coverage (%) | 99.59 | 98.56 | 99.59 | 98.56 |
| Negative positions | 4,674,554 | 4,627,768 | 4,674,554 | 4,627,768 |
| Illumina variants | 1201 | 1162 | 1201 | 1162 |
| ONT variants | 1190 | 1159 | 1196 | 1164 |
| TPs | 1181 | 1155 | 1191 | 1160 |
| FNs | 20 | 7 | 10 | 2 |
| FPs | 9 | 4 | 5 | 4 |
| Sensitivity (%) | 98.33 | 99.40 | 99.17 | 99.83 |
| Precision (%) | 99.24 | 99.65 | 99.58 | 99.66 |
| Jaccard similarity (%) | 97.60 | 99.06 | 98.76 | 99.49 |
| Perfect concordance | 140/157 cases | 149/157 cases | 147/157 cases | 152/157 cases |
| Illumina SNVs | 1194 | 1162 | 1194 | 1162 |
| ONT SNVs | 1180 | 1155 | 1190 | 1160 |
| TPs | 1180 | 1155 | 1190 | 1160 |
| FNs | 14 | 7 | 4 | 2 |
| FPs | 0 | 0 | 0 | 0 |
| Sensitivity (%) | 98.83 | 99.40 | 99.66 | 99.83 |
| Precision (%) | 100 | 100 | 100 | 100 |
| Jaccard similarity (%) | 98.83 | 99.40 | 99.66 | 99.83 |
| Perfect concordance | 145/157 cases | 152/157 cases | 153/157 cases | 155/157 cases |

[a]Blacklist sites are error-prone low-complexity sequences ($n = 15$; 9–42 bp; see text for details).

with 99.17% sensitivity and 99.58% precision by *Nanopolish*, compared to 98.33% sensitivity and 99.24% precision by *Medaka* (Table 2). Undetected variants (false-negatives) were more frequent than erroneous candidates (false-positives), occurring in 14/157 (9%) and 9/157 (6%) of *Medaka* samples, respectively (Supplementary Data 2). Only 1/7 (14%) of consensus indels in the Illumina comparison set was detected by either *Nanopolish* or *Medaka*, while a further five and nine false-positive indels were detected by the respective pipelines (Supplementary Data 2). While the scarcity of consensus indels detected with either sequencing technology prevented a more thorough evaluation of indel accuracy, this indicates that ONT is inadequate for accurate detection of small indels in the SARS-CoV-2 genome. In contrast,

SNVs were detected by *Nanopolish* and *Medaka* with high accuracy: overall, we found 99.66% and 98.83% concordance between ONT and Illumina SNVs, as measured by Jaccard similarity, with identical results in 145/157 (92%) and 153/157 samples (97%), respectively (Table 2).

Inspection of false-positive and false-negative variant candidates detected with ONT sequencing data showed that these tended to occur in low-complexity sequences, which are known to be refractory to ONT base-calling algorithms[23]. For example, false-negative and/or false-positive candidates were found within a 21-bp T-rich site in the *orf1ab* gene in multiple samples (Supplementary Fig. 3a, b). We identified 15 problematic low-complexity sites in the SARS-CoV-2 genome ranging in size from 9 to 42 bp in length that

showed elevated read-level sequencing error rates (Supplementary Fig. 1d and Supplementary File 1). Exclusion of these positions (~1% of the genome) improved the fidelity of ONT variant detection, with consensus SNVs in the Illumina comparison set being detected with 99.83% and 99.40% sensitivity by *Nanopolish* and *Medaka*, respectively, and perfect precision for both. Consensus SNVs detected with the *Nanopolish* workflow were identical between ONT and Illumina data in 155/157 (99%) of samples (Table 2 and Supplementary Data 3). This suggests that the accuracy of nanopore WGS may be improved via the exclusion of a small number of 'blacklist' low-complexity sites in the SARS-CoV-2 genome from downstream analysis.

We next assessed the impact of sequencing depth on ONT performance. To do so, we down-sampled nanopore sequencing reads from a uniform 200-fold coverage across the SARS-CoV-2 genome and repeated variant detection across a range of coverage depths (see Methods section). Both sensitivity and precision of variant detection were strongly influenced by sequencing coverage, showing a sharp decline below ~50-fold coverage depth, with minimal improvement observed above ~60-fold (Supplementary Fig. 1a, b). As above, excluding error-prone low-complexity sequences afforded consistent improvements to sensitivity and overall concordance across the range of depths tested (Fig. 1a, b).

To verify these observations and assess reproducibility, we re-sequenced 12 specimens, selected to cover a range of SARS-CoV-2 titres ($C_t = 10^3 - 10^7$), to generate triplicate ($n = 3$) data on both Illumina and ONT platforms. We measured reproducibility by performing pairwise comparisons of detected variant candidates between replicates for a given sample (Supplementary Data 4). No discordant variants were detected between Illumina replicates across any of the 36 pairwise sample comparisons (309 variants total), confirming the reliability of short-read WGS. ONT also showed high reproducibility, with 99.36% Jaccard similarity between *Medaka* replicates for consensus variants (310 total) and perfect concordance for SNVs (Supplementary Data 4).

In summary, ONT sequencing enabled highly accurate and reproducible detection of consensus-level SNVs in SARS-CoV-2 patient isolates but appears generally unsuitable for the detection of small indel variants.

**Detection of intra-specimen variation.** Within-host genetic diversity is a common feature of RNA viruses, with divergent quasi-species present in a single infection. Within-host diversity may help infecting viruses evade the host immune response, adapt to changing environments and can cause more severe and/or long-lasting disease[28–30]. Resolving this diversity may also better inform studies of virus transmission than consensus-level phylogenetics alone[31–33]. Therefore, we next evaluated the capacity of nanopore sequencing to identify intra-specimen genetic variation by detecting variants present at sub-consensus frequencies (i.e. variants detected in <80% of mapped reads). Analysis of the SARS-CoV-2 synthetic RNA controls and replicate short-read sequencing libraries (see above) showed that sequencing artefacts in Illumina libraries could be misinterpreted as variants at read-count frequencies below ~20% (Supplementary Fig. 2b and Supplementary Data 5), effectively establishing a lower bound for variant detection. We therefore limited our analysis to variants detected at ≥20% frequency, taking variants detected by Illumina sequencing above this level to be genuine. Overall, short-read sequencing identified sub-consensus variants (20–80%) in 54/157 samples, comprising 156 SNVs and 20 indels (Supplementary Data 6).

Using *Varscan2*, we identified 154 sub-consensus SNV candidates in ONT sequencing libraries (Supplementary Data 6). We detected 119 SNVs (sensitivity = 76.3%) in the Illumina

comparison set and 25 false-positives (precision = 82.6%; Supplementary Data 6). Read-count frequencies for variants identified with both technologies were correlated ($R^2 = 0.69$), indicating that these were bona fide variants, rather than sequencing artefacts (Fig. 1c). While the overall performance of sub-consensus SNV detection was quite poor, most false-positives and false-negatives were confined to the lower end of the frequency range assessed here (Fig. 1c, d). For example, SNVs at high (60–80%) and intermediate (40–60%) sub-consensus frequencies were detected with relatively high sensitivity (95.7%, 91.3%) and precision (100%, 97.7%), whereas low-frequency variants (20–40%) were detected with low sensitivity (63.2%) and precision (69.6%; Fig. 1d). Unsurprisingly, the high rate of indel errors in ONT sequencing libraries meant that they were unsuitable for detecting indel diversity, with errors overwhelming true variants (Supplementary Data 6).

In summary, ONT sequencing enabled detection of within-specimen SNVs at frequencies from ~40–80% with adequate accuracy but was generally unsuitable for the detection of indels or rare SNVs (<40%).

**Detection of structural variation.** Large genomic deletions or rearrangements can have a major impact on virus function and evolution, however, there are currently just a few reported cases of SARS-CoV-2 specimens harbouring structural variants (SVs)[15,34]. Therefore, we next evaluated the detection of SVs in SARS-CoV-2 specimens with ONT sequencing. We used *NGMLR-Sniffles* to identify potential SVs in ONT libraries and validated these with supporting evidence from short-read sequencing (see Methods section).

Across all SARS-CoV-2 patient specimens, we detected 16 candidate deletions ranging in size from 15 to 1840 bp (Table 3), while no other SV types were identified. Of these, 13/16 were supported by split short-read alignments and/or discordant read-pairs in matched Illumina libraries (Supplementary Fig. 4a and Table 3). For 7/16 candidates, short-read evidence confirmed the presence of the deletion but indicated that the breakpoint position was not accurately placed by ONT reads (Supplementary Fig. 4b and Table 3). Among the thirteen deletions detected by both platforms were examples in genes *S*, *M*, *N*, *ORF3*, *ORF6*, *ORF8* and *orf1ab* (Table 3). Only one variant, a 328-bp deletion in *ORF8* (Supplementary Fig. S4c), was detected in multiple specimens, although highly similar (but not identical) 28 bp and 29 bp deletions were also detected in *S* in two unrelated specimens (Supplementary Fig. 4d).

Overall, this analysis demonstrates that large deletions can be reliably detected using ONT sequencing and suggests that structural variation in the SARS-CoV-2 genome is more common and diverse than currently appreciated.

**Discussion**
Viral WGS can be used to study the transmission and evolution of SARS-CoV-2, and is increasingly recognised as a critical tool for public health responses to COVID-19. Nanopore sequencing offers an alternative to established short-read platforms for viral WGS with several advantages. ONT devices: (i) are relatively inexpensive, highly portable and require minimal associated laboratory infrastructure; (ii) enable rapid generation of sequencing data and even real-time data analysis; (iii) require comparatively simple procedures for library preparation and; (iv) offer flexibility in sample throughput, accommodating single (e.g., Flongle), multiple (e.g., MinION/GridION) or tens/hundreds (e.g., PromethION) of specimens per flow-cell[16,18]. Therefore, ONT sequencing could further empower SARS-CoV-2 surveillance initiatives by enabling point-of-care WGS analysis and improved

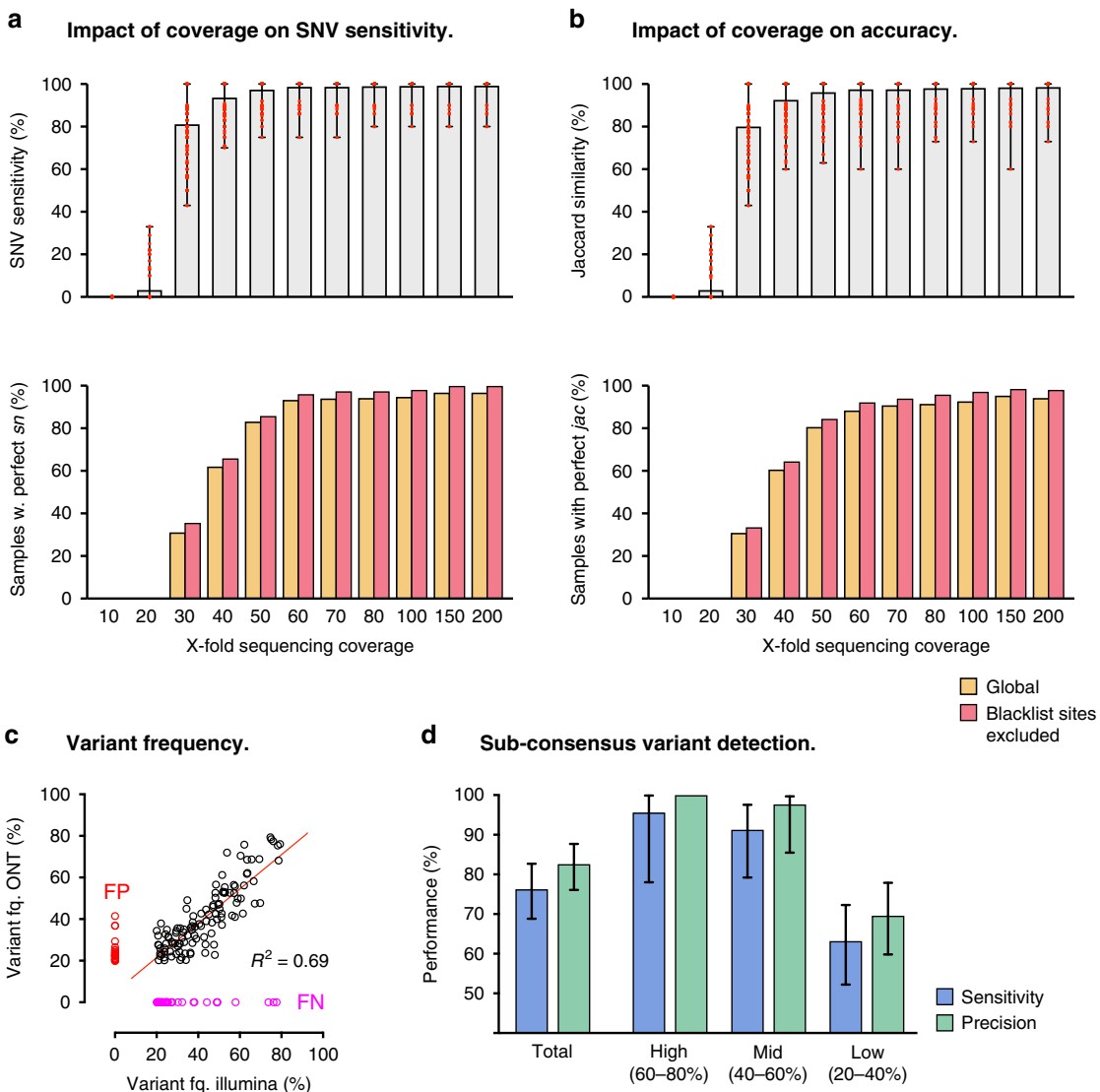

**Fig. 1 Variant detection performance for whole-genome ONT sequencing of SARS-CoV-2.** (**a**; upper) Sensitivity with which Illumina comparison SNVs at consensus-level variant frequencies (80–100%) were detected via ONT sequencing on matched SARS-CoV-2 specimens ($n = 157$). Bars show mean ± range. (**a**; lower) Fraction of specimens tested in which SNVs were detected with perfect sensitivity (*sn*). Data are plotted separately for genome-wide variant detection (gold) and variant detection with error-prone 'blacklist' sites excluded (red). **b** Same as in **a** but Jaccard similarity (*jac*) scores for all variant candidates are plotted instead of SNV *sn*. **c** Correlation of variant frequencies observed for SNV candidates detected at sub-consensus frequencies (20–80%) with Illumina and ONT sequencing. Candidates detected with ONT but not Illumina were considered to be false-positives (FP; red) and candidates detected with Illumina but not ONT were considered to be false-negatives (FP; pink). **d** Sensitivity (blue) and precision (green) of SNV detection with ONT sequencing at sub-consensus variant frequencies (20–80%). Data are plotted separately for high (60–80%), intermediate (40–60%) and low (20–40%) frequencies. Error bars show 95% confidence intervals (Clopper-Pearson) calculated over all specimens ($n = 157$). Source data are provided as Source Data file.

turnaround time for critical cases, particularly in isolated or poorly resourced settings[35].

Due to the relatively low mutation rate observed in SARS-CoV-2[26], accurate sequence determination is vital to correctly define the phylogenetic structure of disease outbreaks. With ONT sequencing known to exhibit higher read-level sequencing error rates than short-read technologies[23–25], reasonable concerns exist about suitability of the technology for SARS-CoV-2 genomics. Moreover, public databases for SARS-CoV-2 data (e.g., GISAID: https://www.gisaid.org/) already contain consensus genome sequences generated via ONT sequencing, potentially confounding investigations that rely on these resources.

The present study resolves these concerns, demonstrating accurate consensus-level SARS-CoV-2 sequence determination

with ONT data. We report that: (i) variants at consensus-level read-count frequencies (80–100%) were detected with >99% sensitivity and >99% precision across 157 SARS-CoV-2-positive specimens, confirming the suitability of ONT sequencing for standard phylogenetic analyses; (ii) high accuracy and reproducibly was achieved by each of two alternative tools for ONT variant detection, with *Nanopolish* showing modest improvements over *Medaka*; (iii) a minimum ~60-fold sequencing depth was required to ensure accurate detection of SNVs, but little or no improvement was achieved above this level; (iv) false-positive and false-negative variants were typically observed at low-complexity sequences, with fidelity improved by excluding these problematic sites; (v) in contrast to consensus SNVs, ONT sequencing performed poorly in the detection of consensus indels or

**Table 3 Detection of structural variation in SARS-CoV-2 specimens with ONT sequencing.**

| Specimen | SV type | Size | Position | Gene | Supporting ONT reads | Short-read evidence | Breakpoint resolution |
|---|---|---|---|---|---|---|---|
| nCoV_077 | Deletion | 15 | 18019-18034 | orf1ab | 94 | Yes | 0, 0 |
| nCoV_087 | Deletion | 1132 | 1082-2214 | orf1ab | 48 | No | – |
| nCoV_088 | Deletion | 34 | 26786-26820 | M | 75 | Yes | 0,0 |
| nCoV_106 | Deletion | 548 | 6004-6552 | orf1ab | 20 | No | – |
| nCoV_125 | Deletion | 27 | 27263-27290 | ORF6 | 20 | Yes | −2, −3 |
| nCoV_183 | Deletion | 15 | 25533-25548 | ORF3 | 41 | Yes | −2, −2 |
| nCoV_214 | Deletion | 29 | 23554-23583 | S | 28 | Yes | +1, +2 |
| nCoV_200 | Deletion | 328 | 27906-28234 | ORF8 | 385 | Yes | 0, 0 |
| nCoV_209 | Deletion | 639 | 2771-3410 | orf1ab | 48 | Yes | 0, 0 |
| nCoV_211 | Deletion | 1840 | 509-2349 | orf1ab | 22 | No | – |
| nCoV_225 | Deletion | 328 | 27906-28234 | ORF8 | 387 | Yes | 0, 0 |
| nCoV_235 | Deletion | 37 | 26783-26820 | M | 21 | Yes | +3, +4 |
| nCoV_249 | Deletion | 702 | 2664-3366 | orf1ab | 52 | Yes | −1, 0 |
| nCoV_164 | Deletion | 588 | 22690-23278 | S | 59 | Yes | +1, +4 |
| nCoV_083 | Deletion | 28 | 23554-23582 | S | 38 | Yes | 0, 0 |
| nCoV_083 | Deletion | 13 | 29478-29491 | N | 36 | Yes | +1, +1 |

low-frequency variants (such variants should therefore be interpreted with caution); (vi) while the high indel error rate in ONT sequencing impedes accurate detection of small indels, long nanopore reads appear well-suited for the detection of large deletions and potentially other structural variants. Although SNVs alone are sufficient for routine phylogenetic analysis, small indels and large structural variants can profoundly impact gene function and are, therefore, of interest to studies of virus evolution and pathogenicity[15].

As the first systematic evaluation of nanopore sequencing for SARS-CoV-2 WGS, this study removes an important barrier to its widespread adoption in the ongoing COVID-19 pandemic. While short-read sequencing platforms remain the gold-standard for high-throughput viral sequencing, the advantages to portability, cost and turnaround-time afforded by nanopore sequencing imply that this emerging technology can serve an important complementary role in local, national and international COVID-19 response strategies.

## Methods

**Synthetic RNA controls**. Synthetic controls used in this study were manufactured by Twist Biosciences and are commercially available (Catalog item 101024). The controls comprise synthetic RNA generated by in vitro transcription (IVT) of the SARS-CoV-2 genome sequence, representing the complete genome in $6 \times \sim 5$ kb continuous sequences. The controls used in this study are identical in sequence to the Wuhan-Hu-1 reference strain (MN908947.3), allowing sequencing artefacts to be readily identified. Synthetic controls were prepared for sequencing via a protocol established by the ARTIC network for viral surveillance (https://artic.network/ncov-2019). Briefly, reverse-transcription was performed on aliquots of synthetic RNA (at $10^6$ copies per μL) using Superscript IV (Thermo Fisher Scientific) with both random hexamers and oligo-dT primers. Prepared cDNA was then amplified using multiplexed PCR with $98 \times \sim 400$ bp amplicons tiling the SARS-CoV-2 genome (ARTIC V3 primer set; see Supplementary Data 7). Amplification was performed with Q5 Hotstart DNA Polymerase (New England Biolabs) with 1.5 μL of cDNA per reaction. PCR products were cleaned using AMPure XP beads (0.8X bead ratio), quantified using a Qubit fluorometer (Thermo Fisher Scientific) and partitioned into separate aliquots for analysis by short-read and nanopore sequencing. We note that it is not possible to amplify the entire SARS-CoV-2 genome in this way, since amplicons that span boundaries of the $6 \times \sim 5$-kb IVT products necessarily fail. Nevertheless, we were able to evaluate ~95% of the SARS-CoV-2 genome sequence.

**SARS-CoV-2 specimens**. SARS-CoV-2-positive extracts from 157 cases, tested at NSW Health Pathology East Serology and Virology Division (SaViD), were retrieved from storage and included in this study. Wherever relevant, ethical regulations for work with human participants with informed consent were observed, with oversight by HREC at South Eastern Sydney Local Health District (SESLHD; 2020/ETH00287). All specimens were nasopharyngeal swabs originating from patients in New South Wales during March–April 2020. Specimens underwent

total nucleic acid extraction using the Roche MagNA Pure DNA and total NA kit on an automated extraction instrument (MagNA pure 96). Reverse-transcription was performed on viral RNA extracts using Superscript IV VILO Master Mix (Thermo Fisher), which contains both random hexamers and oligo-dT primers. Prepared cDNA was then amplified separately with each of $14 \times \sim 2.5$-kb amplicons tiling the SARS-CoV-2 genome, as described elsewhere[6] (see Supplementary Data 7). Amplification was performed with Platinum SuperFi Green PCR Mastermix (Thermo Fisher) with 1.5 μL of cDNA per reaction. PCR products were cleaned using AMPure XP beads (0.8X bead ratio), quantified using PicoGreen dsDNA Assay (Thermo Fisher). All 14x amplicon products from a given sample were then pooled at equal abundance and partitioned into separate aliquots for analysis by short-read and nanopore sequencing. This strategy ensured that any sequence artefacts potentially introduced during reverse-transcription and/or PCR amplification were common to matched ONT/Illumina samples, so would not be interpreted as false-positive/negatives during technology comparison. Technical replicates were generated by reamplification and sequencing of existing RNA extracts (not by re-extraction).

**Short-read sequencing**. Pooled amplicons were prepped for short-read sequencing using the Illumina DNA Prep Kit, according to the manufacturer's protocol. Samples were multiplexed using Nextera DNA CD Indexes and sequenced on an Illumina MiSeq. Within each sequencing lane, a blank sample was also prepared and sequenced, in order to monitor for contamination and/or index swapping between samples. The resulting reads were aligned to the Wuhan-Hu-1 reference genome (MN908947.3) using bwa mem (0.7.12-r1039)[36]. Primer sequences were trimmed from the termini of read alignments using *iVar* (1.0)[37]. Trimmed alignments were converted to pileup format using *samtools mpileup* (v1.9)[38], with anomalous read pairs retained (--count-orphans), base alignment quality disabled (--no-BAQ) and all bases considered, regardless of PHRED quality (--min-BQ 0). Variants were identified using *bcftools call* (v1.9)[38], assuming a ploidy of 1 (--ploidy 1), then filtered for a minimum read depth of 30 and minimum quality of 20. Variants were classified according to their read-count frequencies as consensus (>80% reads supporting the variant) or sub-consensus (20–80%) variants, with the latter further divided into high (60–80%), intermediate (40–60%) or low-frequency (20–40%). Variants at read-count frequencies below 20% were considered to be potentially spurious and excluded on this basis.

**Nanopore sequencing**. ARTIC amplicons (~400 bp) from the synthetic RNA controls were prepared for nanopore using the ONT Native Barcoding Expansion kit (EXP-NBD104). The longer amplicons (~2.5 kb) used on SARS-CoV-2 patient specimens were prepared for nanopore sequencing using the ONT Rapid Barcoding Kit (SQK-RBK004). Both kits were used according to the manufacturer's protocol. Up to 12 samples were multiplexed on a FLO-FLG001, FLO-MIN106D or FLO-PRO002 or flow-cell and sequenced on a GridION X5 or PromethION P24 device, respectively. In addition, a no-template negative control from the PCR amplification step was prepared in parallel and sequenced on each flow-cell (Supplementary Data 8). The *RAMPART* (v1.0.6) software package[39] was used to monitor sequencing performance in real-time, with runs proceeding until a minimum ~200-fold coverage was achieved across all amplicons. At this point, the run was terminated and the flow-cell washed using the ONT Flow Cell Wash kit (EXP-WSH003), allowing re-use in subsequent runs.

The resulting reads were basecalled using *Guppy* (4.0.14) and aligned to the Wuhan-Hu-1 reference genome (MN908947.3) using *minimap2* (2.17-r941)[40]. The ARTIC tool *align_trim* was used to trim primer sequences from the termini of read

alignments and cap sequencing depth at a maximum of 400-fold coverage. Consensus-level variant candidates were identified using each of two workflows developed by ARTIC (https://github.com/artic-network/artic-ncov2019), using *Nanopolish*[41] or *Medaka* (0.11.5) to variants, respectively. Nanopolish variants candidates were filtered directly with the ARTIC *artic_vcf_filter* tool, while *Medaka* candidates were evaluated by *LongShot* (0.4.1)[42] before filtering. Sub-consensus level variant candidates were identified using *Varscan2* (v2.4.3)[43].

**Performance evaluation**. For synthetic RNA controls, read-level quality metrics, such as sequencing error rates, were derived from read alignments using *pysam-stats*, with any bases that differed from the Wuhan-Hu-1 reference sequence considered errors.

The accuracy of variant detection by ONT sequencing was evaluated by comparison to the set of variants identified by Illumina sequencing in matched cases. To ensure consistent representation of variants across calls generated by different programs: (i) multi-allelic variant candidates were separate into individual SNVs/indels using *bcftools norm* (1.9)[38]; (ii) multi-nucleotide variants were decomposed into their simplest set of individual components using *rtg-tools vcfdecompose* (3.10.1) and; (iii) indels at simple repeats were left-aligned using *gatk LeftAlignAndTrimVariants* (4.0.11.0). Variant candidates identified by Illumina/ONT could then be considered concordant based on matching genome position, reference base and alternative base/s. For a given case, variant candidates identified with ONT and Illumina were classified as true-positives (TPs), candidates identified by ONT but not Illumina as false-positives (FPs) and candidates identified by Illumina but not ONT as false-negatives (FNs). The following statistical definitions were used to evaluate results:

$$\text{Sensitivity} = \text{TP}/(\text{TP} + \text{FN}) \quad (1)$$

$$\text{Precision} = \text{TP}/(\text{TP} + \text{FP}) \quad (2)$$

$$\text{Jaccard similarity} = \text{TP}/(\text{TP} + \text{FP} + \text{FN}) \quad (3)$$

**Structural variation**. To identify structural variation, nanopore reads were re-aligned to the Wuhan-Hu-1 reference genome (MN908947.3) using the rearrangement-aware aligner *NGMLR* (v0.2.7)[44]. *Sniffles* (v1.0.11)[44] was then used to detect candidate variants with a minimum length of 10 bp and ≥20 supporting reads. To validate SVs detected with ONT alignments, split short-read alignments and discordant read-pairs were extracted from matched Illumina libraries using *lumpy*[45]. Variant candidates were then manually inspected to verify evidence from ONT and short-reads and assess breakpoint position reFurther information on research design is available in the Nature Research Reporting Summary linked to this article.solution.

**Reporting summary**. Further information on research design is available in the Nature Research Reporting Summary linked to this article.

## Data availability
Raw data has been deposited to the Sequence Read Archive under Bioproject PRJNA675364. Consensus genomes for the majority of samples sequenced here have been deposited separately to GISAID (see Supplementary Data 1). Source data are provided with this paper.

## Code availabilty
Software used in this study is generally open source and all publicly available. Full descriptions, including parameters and version numbers are provided in the Methods section, and further detail on the bioinformatics protocols can be found at: https://github.com/Psy-Fer/SARS-CoV-2_GTG.

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

## Acknowledgements

We acknowledge the following funding support: UNSW COVID-19 Rapid Response Research Initiative (to W.D.R.), MRFF Investigator Grant APP1173594, Cancer Institute NSW Early Career Fellowship 2018/ECF013 and philanthropic support from The Kinghorn Foundation (to I.W.D.).

## Author contributions

Z.N., M.Y., K.W.K., S.S.B., W.D.R. and S.J.vH. oversaw collection and handling of SARS-CoV-2 specimens. T.A., A.V. and I.S. performed cDNA synthesis and PCR amplification of SARS-CoV-2 samples. J.M.H. and I.S. performed ONT library preparation and sequencing. A.G.B. performed Illumina library preparation and sequencing. J.M.F., H.G., I.S., S.J.vH. and I.W.D. performed bioinformatics analysis. I.W.D. prepared the figures. R.A.B., W.D.R., S.J.vH. and I.W.D. prepared the manuscript with support from co-authors.

## Competing interests

H.G. and J.M.F. have previously received travel and accommodation expenses to attend ONT conferences. The authors declare no other competing interests.
