## [Peer Review File · Nature Communications]

REVIEWER COMMENTS

Reviewer #1 (Remarks to the Author):

This is a clearly written, well-described validation study comparing the performance of a ONT sequencing vs Illumina on 157 SARS-CoV-2 clinical samples prepared using a previously described amplicon-based protocol. In addition, the authors report evidence for some intriguing structural variants in a substantial number of clinical SARS-CoV-2 genomes including very large deletions, and smaller deletions, some of which are shared by multiple clinical samples. The ONT platform and its support for long reads would be a good candidate to explore this further, and many researchers performing direct RNA sequencing will no doubt be interested in this part of the report.

A careful validation study is timely and valuable, particularly as ONT sequencing is now being deployed by multiple laboratories for SARS-CoV-2 outbreak tracing, and as such this study has the potential to be highly cited and useful to laboratories looking to develop their sequencing capacity. It is with a view to the importance of this work that the following suggestions are made.

Major comments:

- Short read sequencing is the most commonly adopted, but many labs currently do use ONT already, including for SARS-CoV-2 in public health settings, eg: Meredith, Hamilton et al <https://www.medrxiv.org/content/10.1101/2020.05.08.20095687v1.full.pdf>; Walker et al <https://www.eurosurveillance.org/content/10.2807/1560-7917.ES.2020.25.22.2000746?TRACK=RSS> etc – please consider reviewing at least a couple of the relevant papers in the introduction. Some validation has also been done specifically for amplicons on ONT, eg. https://wwwnc.cdc.gov/eid/article/26/10/20-1800_article
- There are no replicates of clinical samples. This is a major limitation of any validation study: it is not clear to what extent the differences can be attributed to the platform, rather than different sequencing runs on the same samples. There are very few replicate-discordant calls shown in the Twist control replicates, but it is not clear whether this may be due to higher template copy number than in the clinical samples (which would make the Twist controls less vulnerable to errors), or possibly to mapping of reads back to their identical reference (which again makes it more likely that a correct call would be made in Twist, rather than in a randomly taken clinical sample). Even in Twist controls, there are some discrepancies between the ONT replicates shown in Table 1. To determine whether errors in clinical samples are attributable to Illumina/ONT or are due to stochastic run-to-run variation, it would be very important to include a subset of clinical samples sequenced in

duplicate on both platforms, and a comparison of concordance in variants between platforms vs between runs.

- What were the negative controls? There is no mention of these in the methods; negative (template-free) controls are essential for any multiplexed sequencing run, regardless of sequencing platform. Please give the number of mapping (SARS-CoV-2) reads in the negative control samples and indicate which run corresponded to which set of negative controls. Negative controls can be any sample that is expected to contain zero SARS-CoV-2 reads (ie, human RNA, other viral RNA, water control, etc, depending on prep method), which have been treated identically to real SARS-CoV-2 samples. Without negative controls it is not clear whether contamination or index hopping could have contributed to the reported variant calls.

- Perhaps I missed this, but were there any variants consistently missed by ONT, or by Illumina, and if so, were these specific to any genome positions apart from the low-complexity regions shown? In particular, the ends of sub-genomic RNA fragments and the leader-TRS splice sites are vulnerable to sequencing error, for a variety of methodological and biological reasons. What proportion of errors fell within these regions? Would excluding these (similarly to excluding the low-complexity regions) improve variant calling accuracy and concordance?

- Input RNA copy number has a significant impact on the accuracy of variant calls, and on the ability of any method to detect intra-specimen variation. For clinical samples, is it possible to estimate the input RNA quantity using Ct values? For Twist controls, please show the input copy number per reaction, as calculated from the amount of control RNA. Was there anything special about samples with/without intrahost variation - did they vary by Ct?

- “sequencing artefacts in Illumina libraries could be misinterpreted as variants at read-count frequencies below ~20% (Fig. S2b)” – This is an exceptionally high error threshold, and is unlikely to be arising directly from sequencing. Is it possible that correlated mismatch frequencies are arising from IVT, rather than from sequencing error? Or perhaps from human control RNA, if this was mixed with the Twist standard as directed by the manufacturer? It would be a good idea to ask Twist Biosciences to comment, and perhaps request some of their control data for the sample used in this study, to verify whether any of the identified variants were indeed present prior to library prep. In supplementary figure S1, please identify the high-frequency variants and explain which replicates support these, and state which variants are supported by both technologies. Is it possible to also see a similar set of correlation plots for samples that had undergone IVT independently - how well-correlated were the variants?

- Please give genomic coordinates of variants where practical, especially Table 3 (deletions) and in supplementary figures S3 and S4. Some of the reported deletions in table 3 are very large and will no doubt create a lot of interest.
- Data availability: the short-read and ONT data should be made publicly available at publication; please indicate how data will be shared.

Minor comments:

- It would be very helpful to see an overview table for the clinical samples in this study (eg how collected, Ct) and either the phylogenetic lineage for eg. from Pangolin software or a phylogenetic tree in a global context, to better understand the relevance and scope of the dataset.
- How many reads (depth) were required to support an Illumina variant call? Please expand the methods section to explain the full process, eg the following does not explain what 'quality' (base quality, mapping quality?) was used to filter variants, and what process was used to generate the variants from the pileup: "Variants were identified using samtools mpileup (v1.9)32 and filtered for a minimum quality of 20"
- Supplementary figure S2 – legend doesn't seem to match the figure, and doesn't fully describe it.
- How were the low-complexity regions in the BED file determined? Please include this in the methods.

Reviewer #2 (Remarks to the Author):

Dear authors,

The manuscript by Bull and Adikari et al. demonstrates application of nanopore and Illumina sequencing SARS-CoV-2 samples. The authors have used synthetic controls to establish the methods and technologies, followed by analyzing clinical samples. This is good work and I think will be useful to the community.

I think the authors performed the Illumina data analyses reasonably well. I quite liked the comparison of errors as well as the work on short and long indels for SV analysis.

I have a few comments:

1. I am not sure I understood the rationale of using different size amplicons for nanopore sequencing. The authors use 400 bp amplicons on the synthetic RNA (which makes sense given ARTIC protocol usage). However, then then used 2.5 kb amplicons for clinical samples. Why?
2. Following up on point 1, did the authors run the full ARTIC informatics pipeline? How did it compare with the results?
3. My impression is that Guppy 4.0+ has fixed low complexity calling issues that were present in Guppy 3.6. I am curious if the authors have tested it on these data?
4. Biologically, what are the implications of intra-specimen variation? My impression (could be incorrect) is that the general assumption by the community is the presence of a single strain in a patient. Could the authors comment on this?
5. The authors surmise that nanopore sequencing is capable of SNV calling and large SV detection for SARS-CoV-2 samples, but not for short indels. Their analysis backs these findings. For maintaining an epidemiology of the virus, as well as the phylogeny, SNVs are quite sufficient. Could the authors comment on the biological implications of the short indels?
6. The authors claim this is the first systematics evaluation. I would argue that several groups are working on SARS-CoV-2 sequencing using both nanopore and Illumina. This work is meritorious, but I would argue that more substantiation would be required to claim first in field.

My best wishes to the authors.

REVIEWER #1

This is a clearly written, well-described validation study comparing the performance of a ONT sequencing vs Illumina on 157 SARS-CoV-2 clinical samples prepared using a previously described amplicon-based protocol. In addition, the authors report evidence for some intriguing structural variants in a substantial number of clinical SARS-CoV-2 genomes including very large deletions, and smaller deletions, some of which are shared by multiple clinical samples. The ONT platform and its support for long reads would be a good candidate to explore this further, and many researchers performing direct RNA sequencing will no doubt be interested in this part of the report.

A careful validation study is timely and valuable, particularly as ONT sequencing is now being deployed by multiple laboratories for SARS-CoV-2 outbreak tracing, and as such this study has the potential to be highly cited and useful to laboratories looking to develop their sequencing capacity. It is with a view to the importance of this work that the following suggestions are made.

We thank **Reviewer 1** for their careful consideration and constructive feedback on the manuscript. Point-by-point responses to the reviewer's comments are provided below.

1.1. Short read sequencing is the most commonly adopted, but many labs currently do use ONT already, including for SARS-CoV-2 in public health settings, eg: Meredith, Hamilton et al <https://www.medrxiv.org/content/10.1101/2020.05.08.20095687v1.full.pdf>; Walker et al <https://www.eurosurveillance.org/content/10.2807/1560-7917.ES.2020.25.22.2000746?TRACK=RSS> etc – please consider reviewing at least a couple of the relevant papers in the introduction. Some validation has also been done specifically for amplicons on ONT, eg. https://wwwnc.cdc.gov/eid/article/26/10/20-1800_article

We accept that we could have better acknowledged current applications of nanopore sequencing for SARS-CoV-2 research and surveillance. To address this, we have added the following sentence to the introduction and cited several additional articles, including those mentioned above by the reviewer:

Although protocols for ONT sequencing of SARS-CoV-2 have been established and applied in both research and public health settings²⁰⁻²², adoption of the technology has been limited due to concerns around its accuracy.

The validation study that Reviewer 1 mentions (Paden et al 2020), demonstrates the use of an amplicon-based ONT sequencing assay. However, only a single SARS-CoV-2 specimen is analysed in the study, which is insufficient to judge the analytical accuracy of the sequencing assay. Therefore, we maintain the view that our study is the first to *systematically and rigorously* evaluate the analytical performance of ONT SARS-CoV-2 sequencing.

1.2. There are no replicates of clinical samples. This is a major limitation of any validation study: it is not clear to what extent the differences can be attributed to the platform, rather than different sequencing runs on the same samples. There are very few replicate-discordant calls shown in the Twist control replicates, but it is not clear whether this may be due to higher template copy number than in the clinical samples (which would make the Twist controls less vulnerable to errors), or possibly to mapping of reads back to their identical reference (which again makes it more likely that a correct call would be made in Twist, rather than in a randomly taken clinical sample). Even in Twist controls, there are some discrepancies between the ONT replicates shown in Table 1. To determine whether errors in clinical samples are attributable to Illumina/ONT or are due to stochastic run-to-run variation, it would be very important to include a subset of clinical samples sequenced in duplicate on both platforms, and a comparison of concordance in variants between platforms vs between runs.

At the reviewer's request, we have now generated two additional technical replicates ($n = 3$ overall) for twelve of the SARS-CoV-2 specimens originally analysed, on both Illumina and ONT platforms. These additional experiments allow us to further assess performance and (importantly) the reproducibility of both technologies for SARS-CoV-2 WGS. Echoing the high sensitivity and precision reported in the original manuscript, we also observed high reproducibility, as measured by pairwise comparisons between replicates. This data is presented in an additional supplementary table (see **Supplementary Table 4**) and described in the manuscript as follows:

To verify these observations and assess reproducibility, we re-sequenced twelve specimens to generate triplicate ($n = 3$) data on both Illumina and ONT platforms (see **Methods**). We measured reproducibility by performing pairwise comparisons of detected variant candidates between replicates for a given sample (**Supplementary Table 4**). No discordant variants were detected between Illumina replicates across any of the 36 pairwise sample comparisons (309 variants total), confirming the reliability of short-read WGS. ONT also showed high reproducibility, with 99.36% Jaccard similarity between *Medaka* replicates for consensus variants (312 total) and perfect concordance for SNVs (**Supplementary Table 4**).

1.3. What were the negative controls? There is no mention of these in the methods; negative (template-free) controls are essential for any multiplexed sequencing run, regardless of sequencing platform. Please give the number of mapping (SARS-CoV-2) reads in the negative control samples and indicate which run corresponded to which set of negative controls. Negative controls can be any sample that is expected to contain zero SARS-CoV-2 reads (ie, human RNA, other viral RNA, water control, etc, depending on prep method), which have been treated identically to real SARS-CoV-2 samples. Without negative controls it is not clear whether contamination or index hopping could have contributed to the reported variant calls.

We apologise for leaving this information out of the original submission. No-template negative controls were included during the PCR amplification step and these were sequenced on ONT flow cells in parallel to SARS-CoV-2 specimens to assess potential cross-contamination between samples within a given batch, which could occur during amplification, library preparation or via mis-identification of sample barcodes in the resulting reads. Negative controls were also included during Illumina sequencing runs, to assess potential cross-contamination and/or index hopping between samples. In both cases, the rate of cross-contamination to the negative controls was typically low and unlikely to result in spurious detection of variants in specimens analysed, which aligns with our finding of relatively few false-positive variants.

We have now added some explanation about the use of negative controls to the **Materials & Methods** section and, at the reviewer's request, have included an additional supplementary table (see **Supplementary Table 6**) that reports read counts for the negative controls.

For ONT:

Up to twelve samples were multiplexed on a FLO-FLG001, FLO-MIN106D or FLO-PRO002 or flow-cell and sequenced on a GridION X5 or PromethION P24 device, respectively. In addition, a no-template negative control from the PCR amplification step was prepared in parallel and sequenced on each flow-cell.

For Illumina:

Samples were multiplexed using Nextera DNA CD Indexes and sequenced on an Illumina MiSeq. Within each sequencing lane, a blank sample was also prepared and sequenced, in order to monitor for contamination and/or index swapping between samples.

1.4. Perhaps I missed this, but were there any variants consistently missed by ONT, or by Illumina, and if so, were these specific to any genome positions apart from the low-complexity regions shown? In particular, the ends of sub-genomic RNA fragments and the leader-TRS splice sites are vulnerable to sequencing error, for a variety of methodological and biological reasons. What proportion of errors fell within these regions? Would excluding these (similarly to excluding the low-complexity regions) improve variant calling accuracy and concordance?

False-negatives (FNs) and false-positives (FPs) were relatively rare, occurring in 14/157 (9%) and 9/157 (6%) of *Medaka* samples, respectively, which makes it difficult to rigorously investigate the type of effect that the reviewer is describing. We did not identify any FPs or FN near leader-TRS sites. The only discernible pattern was a tendency for FPs and FN to occur within positions of low sequence complexity, as described in the manuscript. For example, multiple FNs occurred at a single T-rich repeat that is shown in **Fig. S3**.

1.5. Input RNA copy number has a significant impact on the accuracy of variant calls, and on the ability of any method to detect intra-specimen variation. For clinical samples, is it possible to estimate the input RNA quantity using Ct values? For Twist controls, please show the input copy number per reaction, as calculated from the amount of control RNA. Was there anything special about samples with/without intrahost variation - did they vary by Ct?

Synthetic SARS-CoV-2 RNA controls from Twist were supplied in solution at a known RNA concentration (10^6 copies per μL). This detail has been added to the **Materials & Methods** section. In order to estimate SARS-CoV-2 RNA copy number in clinical specimens, we have serially diluted this stock and performed qPCR to generate a standard curve

that defines the relationship between SARS-CoV-2 RNA concentration (copies μL) and the measured abundance from qPCR (C_t value):

Figure not included in the manuscript. Standard curve defines relationship between SARS-CoV-2 RNA copy number & measured abundance (C_t).

At the reviewer's request, we have used this standard curve to estimate the input RNA quantity for each sample and included these values in **Supplementary Table 1**. We do not see any obvious relationship between RNA copy number and detection of intra-specimen variation in this study. However, it should be noted that all samples used in this study have relatively high viral titre, all having C_t values below 30. Perhaps if samples with lower titre were analysed, such a trend may emerge, but this is beyond the scope of the present study.

1.6. "sequencing artefacts in Illumina libraries could be misinterpreted as variants at read-count frequencies below ~20% (Fig. S2b)" – This is an exceptionally high error threshold, and is unlikely to be arising directly from sequencing. Is it possible that correlated mismatch frequencies are arising from IVT, rather than from sequencing error? Or perhaps from human control RNA, if this was mixed with the Twist standard as directed by the manufacturer? It would be a good idea to ask Twist Biosciences to comment, and perhaps request some of their control data for the sample used in this study, to verify whether any of the identified variants were indeed present prior to library prep. In supplementary figure S1, please identify the high-frequency variants and explain which replicates support these, and state which variants are supported by both technologies. Is it possible to also see a similar set of correlation plots for samples that had undergone IVT independently - how well-correlated were the variants?

We direct the reviewer to **Fig. S2b**, which shows read-count frequencies for erroneous candidates detected by short-read sequencing of synthetic RNA controls from Twist:

Fig. S2b. Read-count frequencies of illumina FPs detected in synthetic RNA controls.

The figure shows that almost all errors are restricted to <10% frequency and most are <5%, which will likely not surprise the reviewer, who suggests that 20% is an unnecessarily high threshold to use here. In fact, just a single substitution error in each of 2 (out of 8) replicates was detected at 20% frequency. Therefore, in a real-world scenario, 5% or 10% would likely be a preferable cut-off for detecting sub-consensus variants with short-read sequencing. However, we chose to use 20% as the cut-off for further analysis as this was the minimum cut-off that excluded **ALL** false-positives. We acknowledge that this is very conservative and somewhat arbitrary but, in a benchmarking study of this nature, we believe it is important to err on the side of caution in establishing a reliable 'ground-truth' set of variants.

Anyhow, the choice of cut-off is also largely inconsequential for our purposes here, since ONT sequencing is unable to detect sub-consensus variants with adequate sensitivity and precision even in the 20-40% frequency range (see **Fig. 1d**). Therefore, while we partially agree with the reviewer's view, we have decided not to change this cut-off from the 20% used in the original manuscript.

1.7. Please give genomic coordinates of variants where practical, especially Table 3 (deletions) and in supplementary figures S3 and S4. Some of the reported deletions in table 3 are very large and will no doubt create a lot of interest.

As requested, we have added genomic coordinates for the large deletions reported in **Table 3** and for variants/regions depicted in **Fig. S3** and **Fig. S4**.

1.8. Data availability: the short-read and ONT data should be made publicly available at publication; please indicate how data will be shared.

All data will be publicly available at publication. See **Data Availability** statement in the manuscript for details.

1.9. It would be very helpful to see an overview table for the clinical samples in this study (eg how collected, Ct) and either the phylogenetic lineage for eg. from Pangolin software or a phylogenetic tree in a global context, to better understand the relevance and scope of the dataset.

As requested, we have added further details describing SARS-CoV-2 specimens to **Supplementary Table 1**, including collection date, measured SARS-CoV-2 RNA abundance (C_t), estimated SARS-CoV-2 RNA concentration (cp/uL) and Pangolin lineages.

1.10. How many reads (depth) were required to support an Illumina variant call? Please expand the methods section to explain the full process, eg the following does not explain what 'quality' (base quality, mapping quality?) was used to filter variants, and what process was used to generate the variants from the pileup: "Variants were identified using samtools mpileup (v1.9)³² and filtered for a minimum quality of 20"

As requested, we have provided further detail in the **Materials & Methods** section to describe the detection and filtering of variants in short-read sequencing libraries:

The resulting reads were aligned to the Wuhan-Hu-1 reference genome (MN908947.3) using *bwa mem* (0.7.12-r1039)³⁶. Primer sequences were trimmed from the termini of read alignments using *iVar* (1.0)³⁷. Trimmed alignments were converted to pileup format using *samtools mpileup* (v1.9)³⁸, with anomalous read pairs retained (--count-orphans), base alignment quality disabled (-no-BAQ) and all bases considered, regardless of PHRED quality (--min-BQ 0). Variants were identified using *bcftools call* (v1.9)³⁸, assuming a ploidy of 1 (--ploidy 1), then filtered for a minimum read depth of 30 and minimum quality of 20. Variants were classified according to their read-count frequencies as consensus (>80% reads supporting the variant) or sub-consensus (20-80%) variants, with the latter further divided into high (60-80%), intermediate (40-60%) or low-frequency (20-40%). Variants at read-count frequencies below 20% were considered to be potentially spurious and excluded on this basis.

1.11. Supplementary figure S2 – legend doesn't seem to match the figure, and doesn't fully describe it.

We thank the reviewer for pointing this out and have fixed the legend for **Fig. S2**.

REVIEWER #2

The manuscript by Bull and Adikari et al. demonstrates application of nanopore and Illumina sequencing SARS-CoV-2 samples. The authors have used synthetic controls to establish the methods and technologies, followed by analyzing clinical samples. This is good work and I think will be useful to the community. I think the authors performed the Illumina data analyses reasonably well. I quite liked the comparison of errors as well as the work on short and long indels for SV analysis.

We thank **Reviewer 2** for their careful consideration and constructive feedback on the manuscript. Point-by-point responses to the reviewer's comments are provided below.

2.1. I am not sure I understood the rationale of using different size amplicons for nanopore sequencing. The authors use 400 bp amplicons on the synthetic RNA (which makes sense given ARTIC protocol usage). However, then then used 2.5 kb amplicons for clinical samples. Why?

Early in the pandemic, our team adopted the custom 14 x ~2.5 kb amplicon scheme for SARS-CoV-2 WGS described in the article. We prefer this approach to the popular 98 x ~400 bp scheme from ARTIC because it provides more even genome coverage, lower chance of primer-site mutations and superior detection of structural variants, given the higher read-lengths achieved. All SARS-CoV-2 patient specimens that come through our lab (included those presented in the present manuscript) are analysed using this amplicon scheme.

The reviewer has correctly noted that we chose to analyse the synthetic RNA controls from Twist using the ARTIC amplicon scheme, rather than our preferred custom scheme. There is a simple, practical explanation for this choice. The synthetic RNA controls do not represent the entire ~30 kb SARS-CoV-2 genome in single continuous RNA molecules but are expressed via IVT in 5 x ~6 kb segments. Any amplicon that spans the boundary between two adjoining segments cannot be amplified; meaning that 5 x amplicons will necessarily fail. For our 14 x ~2.5kb amplicon scheme, this means ~36% (5/14) of the genome cannot be analysed. Because the ARTIC amplicons are smaller, a smaller fraction (5/98 = ~5%) of the genome is impacted by this boundary affect. Therefore, to maximise the fraction of the genome that is analysable in the synthetic RNA controls, we chose to use the ARTIC scheme for these experiments.

We do not feel it is necessary to include this rather technical explanation in the article, but we hope it has addressed the reviewer's query.

2.2. Following up on point 1, did the authors run the full ARTIC informatics pipeline? How did it compare with the results?

Yes, we have used the full ARTIC bioinformatics pipeline for analysis of all ONT data in the study. However, the pipeline we used in the original submission is no longer the most up-to-date version (v1.1.1.). Additionally, we only used one of two alternative ARTIC pipelines; this being the one that employs *Medaka* + *Longshot* for variant detection, and ignored the alternative pipeline that instead employs *Nanopolish*.

While the pipeline used in our original manuscript submission performed well, we believe a rigorous comparison of these alternative bioinformatics workflows is valuable to the community and have therefore re-analysed all data using both versions of the latest ARTIC pipeline (*ARTIC-Medaka* vs *ARTIC-Nanopolish*). Note that, in all cases, we also used an updated version of Guppy (v4.0.14), as requested below (see **2.3**).

All data presented in the article has been updated accordingly and a clear comparison of the alternative pipelines is presented in **Table 2**, **Supplementary Table 3** and described within the **Results** and **Discussion**. Overall, *Nanopolish* performs slightly better than *Medaka*, which we believe is a result worth reporting.

2.3. My impression is that Guppy 4.0+ has fixed low complexity calling issues that were present in Guppy 3.6. I am curious if the authors have tested it on these data?

As requested, we have reanalysed all samples using an updated version of Guppy (from 3.6 to 4.0.14). We have also reanalysed the data using the latest versions of the ARTIC bioinformatics pipeline, with both *Nanopolish* and *Medaka*, in order to provide a rigorous comparison of these bioinformatic variables (see **2.2**).

As the reviewer suggests, these updates have improved the overall accuracy of variant detection with ONT data, with just a handful of FNs (both SNVs and indels) and FPs (only indels) remaining across the dataset. The main challenge still lies in the detection of short indels, as we mention throughout.

2.4. Biologically, what are the implications of intra-specimen variation? My impression (could be incorrect) is that the general assumption by the community is the presence of a single strain in a patient. Could the authors comment on this?

Within-host genetic variation is commonly observed among RNA viruses and, therefore, an expected feature of SARS-CoV-2 infections. There can be multiple different strains infecting a single individual and further divergence of 'quasi-species' within the host may occur during the period of infection. At least one study has reported extensive within-host genetic variation in SARS-CoV-2 patients (see Lythgoe et al 2020). The frequency of within-host diversity is also likely to increase over time, as the diversity of circulating strains increases, especially in emergent strains carrying mutations in RdRp and other genes of replication that cause elevated mutation rates (see Pachetti et al 2020).

The detection of within-host genetic diversity is interesting for several reasons:

1. The generation of within-host diversity can help infecting viruses to evade host immune responses or adapt to changing environments (see Hensley et al 2009, Henn et al 2012) and may, therefore, be an obstacle in development of vaccines and/or antibody-based therapeutics.
2. Evidence from other viruses suggests that infection by diverse viral populations leads to more severe and/or long-lasting infections (see Vignuzzi et al 2005, Stern et al 2017).
3. Inclusion of minor variants / virotypes (ie variants at sub-consensus frequencies) can improve the resolution of phylogenetic analysis, potentially resolving epidemiological clusters that are indistinguishable at the level of consensus variants (see Worby et al 2017, De Maio et al 2018, Wymant et al 2018).

As requested, we have elaborated further on the relevance of within-host diversity in SARS-CoV-2 within the article:

Within-host genetic diversity is a common feature of RNA viruses, with divergent quasi-species present in a single infection. Within-host diversity may help infecting viruses evade the host immune response, adapt to changing environments and can cause more severe and/or long-lasting disease^{28,29,30}. Resolving this diversity may also better inform studies of virus transmission than consensus-level phylogenetics alone³¹⁻³³.

2.5. The authors surmise that nanopore sequencing is capable of SNV calling and large SV detection for SARS-CoV-2 samples, but not for short indels. Their analysis backs these findings. For maintaining an epidemiology of the virus, as well as the phylogeny, SNVs are quite sufficient. Could the authors comment on the biological implications of the short indels?

The reviewer is correct that SNVs are sufficient for phylogenetic analysis to infer transmission patterns and disease clusters. Small indels and large structural variants are not used in these analyses. However, both small and large indels are common features of RNA viruses and, by their nature, these mutations are more likely to impact gene function than SNVs. While large SVs can see whole ORFs deleted, truncated or rearranged, small indels often cause frameshifts with deleterious impact on gene function. Loss or alteration of gene function can be an important mode of virus evolution. The most prominent example for SARS-CoV-2 is the recent report of a circulating strain carrying a 382 bp near-complete deletion of *ORF8* that exhibits reduced immunogenicity (see Gong et al 2020, Su et al 2020, Young et al 2020). Although we are not aware of any similar strains harbouring small indel mutations, their potential emergence should not be ignored.

As requested, we have elaborated further on the relevance of small and large indels in SARS-CoV-2 within the article:

Although SNVs alone are sufficient for routine phylogenetic analysis, small indels and large structural variants can profoundly impact gene function and are, therefore, of interest to studies of virus evolution and pathogenicity¹⁵.

2.6. The authors claim this is the first systematic evaluation. I would argue that several groups are working on SARS-CoV-2 sequencing using both nanopore and Illumina. This work is meritorious, but I would argue that more substantiation would be required to claim first in field.

We are aware that there are other researchers who have used both ONT and Illumina platforms for SARS-CoV-2 genome sequencing and we suspect that these groups are aware of the advantages and shortcomings of each technology. However, if these groups have performed extensive comparisons on matched samples to evaluate ONT performance, then this data has not yet been published.

There are some studies where direct comparisons on matched samples were reported, such as Paden et al (1 specimen) and Walker et al (11 specimens), and we have cited these in our re-submitted manuscript. However, these sample numbers are insufficient to properly assess the accuracy of ONT sequencing. Due to the low mutation rate of SARS-CoV-2, it is necessary to consider a large number of samples to establish analytical validity. For this reason, we chose to analyse a large cohort of 157 matched samples, encompassing ~1200 polymorphic sites.

Therefore, while we do not claim that we are the first group to perform both ONT and Illumina sequencing on SARS-CoV-2, we stand by the assertion that our study is the first *systematic and rigorous evaluation* of ONT performance. Our study demonstrates that ONT platforms can accurately determine the sequence of SARS-CoV-2, dispelling widespread concerns about the technology and facilitating its further adoption within research and public health initiatives.

REVIEWER COMMENTS

Reviewer #1 (Remarks to the Author):

I would like to thank the authors for their detailed responses and for the additional work done to address the points raised in my review. The responses are thorough and strengthen the paper. I have a few minor outstanding concerns:

Replication:

* Were the 12 new replicates prepared from new RNA aliquots, or by resequencing the existing libraries? I would assume from RNA as this is the only appropriate form of replication for this purpose, but please specify this in the methods. I could not find any additions to the methods specifically about replicates, so I wasn't sure what citing Methods in this new section refers to.

* How were the samples to be resequenced selected? It would be useful to copy across the viral load estimates for these from supplementary table 1 to supp table 4, and also to highlight them in table 1. It is excellent that some lower-VL samples were included for replication; I would urge the authors to highlight this point in the main text as it lends additional weight to their findings.

* I would encourage the authors, at their discretion, to include the new standard curve in the supplementary data as it helps to interpret the tables.

* The 12 replicates are compared only at the majority/consensus variant level. Are the minority calls in the illumina data likewise concordant? If so, does this affect the choice of variant calling threshold?

The following part of my original comments does not seem to have been addressed; please could you indicate your responses:

“It would be a good idea to ask Twist Biosciences to comment, and perhaps request some of their control data for the sample used in this study, to verify whether any of the identified variants were indeed present prior to library prep. In supplementary figure S1, please identify the high-frequency variants and explain which replicates support these, and state which variants are supported by both technologies.”

* Please comment on how many variants were expected in Twist. Due to IVT, there may well be some genuine variants present in the supplier's RNA.

* Please also give the genomic positions of the high-frequency variants you identified, or any variants thought to be genuinely present in the Twist RNA.

* Figure S3 legend - typo in the newly added genome coordinates; 11086-1186 perhaps meant to be 11086-11186?

* Newly added sample description table: this is very useful; I would encourage the authors to add the median Ct value and Ct range of the samples, and the fact that they included A and B lineages, to the main text.

Reviewer #2 (Remarks to the Author):

Dear authors,

Thank you for addressing my comments. I do not have any further questions, and send my best wishes.

Cheers.

REVIEWER #1

I would like to thank the authors for their detailed responses and for the additional work done to address the points raised in my review. The responses are thorough and strengthen the paper.

We thank Reviewer #1 for their thorough evaluation of the study and constructive feedback throughout the process; the revisions suggested have strengthened the manuscript. Point-by-point responses to the reviewer's new comments are detailed below.

1.1. Were the 12 new replicates prepared from new RNA aliquots, or by resequencing the existing libraries? I would assume from RNA as this is the only appropriate form of replication for this purpose, but please specify this in the methods. I could not find any additions to the methods specifically about replicates, so I wasn't sure what citing Methods in this new section refers to.

Replicates were prepared from new aliquots of existing RNA extracts (note that RNA was not re-extracted from the original patient specimens). We have clarified this detail in the **Materials & Methods** section:

Technical replicates were generated by reamplification and sequencing of existing RNA extracts (not by re-extraction).

1.2. How were the samples to be resequenced selected? It would be useful to copy across the viral load estimates for these from supplementary table 1 to supp table 4, and also to highlight them in table 1. It is excellent that some lower-VL samples were included for replication; I would urge the authors to highlight this point in the main text as it lends additional weight to their findings.

Replicate samples were selected based on the availability of surplus RNA from original extractions (since we did not repeat the extractions) and with the intention of covering a wide range of viral titres, although we note that this does not affect the outcomes or interpretation of the experiment.

At the reviewer's request we have added C_t values and estimated SARS-CoV-2 RNA concentrations to **Supplementary Table 4** and added the following note to the main text:

To verify these observations and assess reproducibility, we re-sequenced twelve specimens, selected to cover a range of SARS-CoV-2 titres ($C_t = 10^3$ - 10^7), to generate triplicate ($n = 3$) data on both Illumina and ONT platforms.

1.3. I would encourage the authors, at their discretion, to include the new standard curve in the supplementary data as it helps to interpret the tables.

At the reviewer's request, we have added the standard curve used to estimate SARS-CoV-2 viral titre to the **Supplementary Materials** section (**Fig. S5**).

1.4. The 12 replicates are compared only at the majority/consensus variant level. Are the minority calls in the illumina data likewise concordant? If so, does this affect the choice of variant calling threshold?

We also assessed the concordance of sub-consensus variants among the 12 replicate libraries, but due to relatively low numbers of variants under consideration we prefer not to report this in the manuscript. Across all pairwise comparisons between Illumina replicates (36 comparisons total) we observed no discordant variants within the VAF range 60-80% or 40-60%, 2 discordant indels (no discordant SNVs) within the VAF range 20-40% and 8 discordant variants (both SNVs and indels) in the VAF range 10-20%:

VAF range	concordant variants	discordant variants	concordant SNVs	discordant SNVs	concordant Indels	discordant Indels
60-80%	6	0	6	0	0	0
40-60%	12	0	12	0	0	0
20-40%	20	2	20	0	0	2
10-20%	59	8	59	2	0	6

These results are consistent with our choice to use >20% as our VAF cut-off for sub-consensus variants analysis, as determined by analysis of synthetic RNA controls. Given that they do not support any modification to this cut-off, we prefer not to report this data in the article, but can do so at the Editor's request.

1.5 The following part of my original comments does not seem to have been addressed; please could you indicate your responses:

“It would be a good idea to ask Twist Biosciences to comment, and perhaps request some of their control data for the sample used in this study, to verify whether any of the identified variants were indeed present prior to library prep. In supplementary figure S1, please identify the high-frequency variants and explain which replicates support these, and state which variants are supported by both technologies.”

* Please comment on how many variants were expected in Twist. Due to IVT, there may well be some genuine variants present in the supplier's RNA.

The synthetic RNA controls from Twist are a commercial product and we do not have access to any internal data beyond what is available in the product specification sheets. It is possible that there could be some errors incorporated during their production by IVT, however, if this were the case we would likely detect these errors in multiple replicates from one or both technologies (which we did not). By analysing a large number of technical replicates for the Twist samples ($n = 8$ for both ONT and Illumina), we effectively minimised the risk of any impact from IVT errors.

1.6. Please also give the genomic positions of the high-frequency variants you identified, or any variants thought to be genuinely present in the Twist RNA.

The only high frequency variants detected across multiple replicates were those annotated in the product specification sheets:

<https://www.twistbioscience.com/resources/product-sheet/twist-respiratory-virus-controls>

1.7. Figure S3 legend - typo in the newly added genome coordinates; 11086-1186 perhaps meant to be 11086-11186?

Thanks for picking this up. We have corrected the typo (it should read: 11066-11086).

1.8. Newly added sample description table: this is very useful; I would encourage the authors to add the median Ct value and Ct range of the samples, and the fact that they included A and B lineages, to the main text.

As suggested, we have added the following note to the main text:

Selected specimens covered a range of SARS-CoV-2 lineages and viral titres ($\sim 10^3$ - 10^8 copies/ μ L; **Supplementary Table 1**).